# Single Cell Analysis of Stored Red Blood Cells Using Ultra-High Throughput Holographic Cytometry

**DOI:** 10.3390/cells10092455

**Published:** 2021-09-17

**Authors:** Han-Sang Park, Hillel Price, Silvia Ceballos, Jen-Tsan Chi, Adam Wax

**Affiliations:** 1Department of Biomedical Engineering, Duke University, Durham, NC 27708, USA; hansang.park36@gmail.com (H.-S.P.); hillel.price@duke.edu (H.P.); sissi@email.unc.edu (S.C.); 2Department of Molecular Genetics and Microbiology, Duke University, Durham, NC 27708, USA; jentsan.chi@duke.edu; 3Duke Center for Genomic and Computational Biology, Duke University, Durham, NC 27708, USA

**Keywords:** high throughput cell screening, quantitative phase imaging, red blood cell storage

## Abstract

Holographic cytometry is introduced as an ultra-high throughput implementation of quantitative phase imaging of single cells flowing through parallel microfluidic channels. Here, the approach was applied for characterizing the morphology of individual red blood cells during storage under regular blood bank conditions. Samples from five blood donors were examined, over 100,000 cells examined for each, at three time points. The approach allows high-throughput phase imaging of a large number of cells, greatly extending our ability to study cellular phenotypes using individual cell images. Holographic cytology images can provide measurements of multiple physical traits of the cells, including optical volume and area, which are observed to consistently change over the storage time. In addition, the large volume of cell imaging data can serve as training data for machine-learning algorithms. For the study here, logistic regression was used to classify the cells according to the storage time points. The analysis showed that at least 5000 cells are needed to ensure accuracy of the classifiers. Overall, results showed the potential of holographic cytometry as a diagnostic tool.

## 1. Introduction

Flow cytometry is a powerful tool that measures multiple physical parameters of complex populations of flowing cells over a short period of time [1]. Since its development as a primarily cell-size measuring instrument, flow cytometry has evolved into a sophisticated modality that can simultaneously a wide range of optical and fluorescence parameters [2]. As a diagnostic tool, flow cytometry can be used to obtain information about the biochemical, biophysical, and molecular aspects of broad populations at the single-cell level. Structural and morphological characteristics of cells can be derived from scattered light measurements while fluorescence emission is used to provide molecular information by estimating the number of fluorescent probes bound to various cellular components [3].

As an extension, imaging flow cytometry has become a valuable tool in the past decade that enables greater analysis of cellular morphology to provide additional information to biological studies and clinical diagnosis [4,5,6]. In-depth imagery of individual cells can be used to verify the parameters measured from conventional flow cytometry by obtaining information such as the detailed shape of the cells and the location of labeled biomolecules within them [7,8,9]. Additionally, false positive occurrences can be reduced by analyzing cell images to eliminate non-cell objects such as debris and clusters of cells [10,11,12]. Therefore, users can be more definitive about the outcome of flow cytometry analysis by having access to sample attributes within the cell images.

Another tool that has been recently developed for cell imaging is quantitative phase microscopy (QPM), which can characterize biological cells by their morphological and spectral features with nanometer levels of precision [13,14,15,16]. Currently, the number of cells analyzed by QPM is greatly limited by the need for manual manipulation of the sample stage to find a good field of view. To translate the technique to be a diagnostically significant screening tool, it is necessary to increase the throughput of the system to reach levels similar to those of imaging flow cytometry, which ranges from 1000 to 100,000 cells/s [5]. Therefore, further development of QPM as a diagnostic approach requires the throughput of the system to be significantly increased.

To demonstrate the potential utility of a high throughput QPM method such as holographic cytology, the technique was used to assess the morphological changes of red blood cells (RBCs) that occur during storage, often termed storage lesion. The number of red blood cells that can be imaged with this system is hundreds of times greater than previous studies which have examined RBCs with QPM [15]. Characteristics of storage lesion include morphological, rheological, and biochemical changes to RBCs which reduce their lifetime in circulation [17]. The change in RBCs during storage is of interest since there is an increase in patient risks with transfusion of older blood, and thus significant efforts have been made to improve the viability of stored RBCs [18]. In addition, the stored RBCs are used to boost the hemoglobin and increase athletic performance in blood doping. Currently, however, there is no way to effectively measure the viability of RBCs at the cellular level with high enough throughput to be useful for evaluating units for transfusion. Instead, the clinical standard is to depend on the observed percentage of hemolysis as an assay, neglecting the significant changes in RBC morphology and mechanical properties which can reflect their reduced viability.

In this article, holographic cytometry was presented as a quantitative phase imaging system that can acquire a large number of images of cells in flow by incorporating customized microfluidic chips and stroboscopic illumination. The approach was used to acquire quantitative phase images of a broad population of individual RBCs at different storage time points. To classify these cells, a set of morphological parameters were extracted from each cell image to enable characterization of millions of cells at the single cell level. The extracted physical parameters showed consistent changes over time and suggest new avenues for improved understanding of the effects of storing RBCs. Further, the extracted parameters were used to train a supervised learning algorithm, based on logistic regression, to classify the cells according to their storage time points.

## 2. Materials and Methods

### 2.1. Blood Collection and Preparation

Whole blood from five different healthy donors (Table 1) were collected into CPD-OPTISOL (AS-5) collection sets.

Each unit of whole blood was stored at 1–6 °C for up to 6 weeks for the experiment. The stored blood is removed from the units at specified time points using a sterile-docking system with a fitted valve. The removed blood was centrifuged at 1000 rpm for 5 min to isolate the RBC pellets which were resuspended as a 0.8% hematocrit solution in 5 mL of high refractive index medium (RI = 1.372 at 23 °C). The mixture is then loaded onto a syringe pump system flowing at 3 µL/min for the experiment.

### 2.2. Development of Microfluidic Channels

The following experiments were performed using PDMS (polydimethylsiloxane) channels [19]. The mask design is shown in Figure 1. Since the width of the channels, 40 μm, was much larger than the average diameter of the RBCs, the passages were free from blockages that could potentially disrupt the uniform flow across the channels. In addition, the height of the SU-8 master mold, 5.39 ± 0.18 μm, is comparable to the thickness of the RBCs.

### 2.3. Holographic Cytometry

The RBCs flowing through the channels were imaged using the holographic cytometry shown in Figure 2.

The illumination beam from a laser source (λ_0_ = 640 nm, Δλ = 0.7 nm) was modulated at 300 Hz using an acousto-optic modulator (AOM). The AOM was triggered to synchronize with the frame rate of the camera using an Arduino based microcontroller. The pulse width of the modulated beam is 350 μs and the spatial noise of the system is observed to stabilize at the 1 nm rms level, with an intensity that is lower than the saturation level of the detector. The pulsed illumination beam provides a stroboscopic effect such that the motion of the cells, which were flowing at a maximum velocity of 2.65 mm/s, was effectively frozen during the exposure. Over the duration of one individual pulse of light, the RBCs travelling at this maximum velocity moved only 0.93 μm, which is less than the spatial resolution of the system (0.98 μm), and therefore motion artifacts are minimized.

The output beam from the AOM was coupled into a 1 × 2 fiber coupler which splits the beam into the sample and the reference arms for the off-axis Mach-Zehnder interferometer. The collimated beam from the sample arm passed through the sample and the image was magnified by 33× using an objective lens (20×, 0.4NA). The magnified beam was combined with the collimated reference beam at an angle using a beam splitter to create an off-axis interferogram which was captured by the camera (Teledyne Dalsa, Waterloo, ON, Canada. 4096 × 96 p×, 300 fps). For each sample, 99 sets of 10,000 frames (~33 s each set) were captured at each storage time point.

The interferograms were post-processed frame-by-frame to obtain optical phase delays, ∆*ϕ*, of the wavefronts propagating through the sample, which depend upon refractive index as:(1)$$\Delta \phi \left(x,y,t\right)=\frac{2\pi }{\lambda }\;\left(n_{cell}\left(x,y,t\right)-n_{media}\right)\cdot h\left(x,y,t\right)$$
where *n* is the refractive index, *h* is the height of the sample, and *λ* refers to the wavelength. An example of a phase image sequence of stored RBCs flowing through the channels is shown in Figure 3.

Briefly, the complex field information describing the flowing RBC was retrieved from the interferograms [16]. Each complex field image was digitally refocused by propagating the field to the plane of best focus, as determined by the minimum variance of amplitude of the cells in the first 3 frames [20]. The cells in the subsequent frames of the flow sequences were digitally refocused using the same distance determined in the first 3 frames since the axial positions are restricted during flow by the height of the customized channel, which is comparable to the cell thicknesses [21]. After background subtraction, any remaining noise was removed using a 3rd-order polynomial fit to the field of view excluding the cells. Then, each RBC was segmented from the image using an area and thickness threshold.

Multiple images of the same cell were identified using a modified tracking code from the Computer Vision Toolbox in Matlab [22]. The tracking code first detects moving objects from a given FOV. Then, it predicts the subsequent location of the object based on the previously observed motion using a Kalman filter. The predicted locations across the frames are used to form motion tracks which identify the moving copies of each single cell object. This ensures that identified copies of multiple images of the same cell do not account for multiple counts in the total number of cells imaged by the system.

Since the RBCs are isolated from the background based only on two morphological parameters, another quality check was performed to ensure that non-RBC objects were not included in the analysis, as shown in Figure 4 below.

As can be seen in Figure 4A, there are groups of segmented objects that do not exhibit morphological features that correspond to the physical properties of RBCs based on size and dynamic structure. Therefore, objects with areas below the threshold of 21 μm^2^ as well as those with standard deviation of phase below 0.09 were categorized as too small to be RBC’s. Typical images of these rejected cells are shown in Figure 4C,D. These excluded objects may include platelets or cell fragments which were omitted from further analysis. In addition, some of the segmented objects are multiple cells clumped together during flow as shown in Figure 4B. The clumped cells that are selected based on the circularity (circularity < 0.85 or >1.23) and optical volume (OV > 5 fL) are too large to be single RBC’s and are also excluded from further analysis (typical images shown in Figure 4E).

The set of morphological parameters used in our previous study [15] and 2 additional features, solidity and circularity, were extracted for each RBC using the segmented images. Solidity is the ratio between the area of the cell and the convex hull, with area defined by the bounding region that connects the outer points of the cell [23]. Circularity is the ratio between the perimeter and the area of the cell that should resemble a circle for values close to 1. A previous study with QPI examined optical volume for stored RBC’s and correlated QPI findings with chemically aged cells [21].

## 3. Results

### 3.1. Number of Individual Cells Imaged

The total number of unique, single RBCs imaged with the system for all samples on day 1, 15, and 29, after excluding non-RBC objects, was 9,437,349 cells as shown in Table 2 below.

The total imaging time for each sample on a given day is ~3300 s, comprising 99 sets, 10,000 frames per set, and at a rate of 300 fps. As can be seen in the table above, the throughput of the system ranges from 36 cells/s for Sample D at day 15 to 420 cells/s for Sample B at day 15. The number of RBCs imaged with the system for each storage day and sample is much greater, by a factor of ~300× or higher, than the total number of RBCs imaged in our previous studies of RBCs using translatable sample stages [15]. The capability to image such a large number of cells will provide a more accurate representation of the total population within the storage units than possible with other methods.

### 3.2. Characterization of Morphological Changes over Time

Out of the 25 morphological parameters examined here, optical volume and area showed the most consistent changes over storage time across most of the samples. The average optical volume over the storage time for all samples are shown as line plots in Figure 5A.

As can be seen in the line plots, the optical volume consistently decreased over the storage time for all samples except Sample A, which declined at week 3, but paradoxically rose at the week 5 time point. The change in the parameter over the storage period is also illustrated using the normalized histograms in Figure 5B where the optical volume of the cells decreased over the storage period for Samples B–E.

Similarly, the area of the cells follows the trend as shown by the mean value of this parameter with line plots in Figure 6A.

Generally, the area of the cells also decreased over time, except for Sample A, at the final time point, week 5. The change in this parameter over the storage period is also shown using normalized histograms in Figure 6B. It should be noted that the decrease in cell area for Samples B–E may occur at different time points. For Samples B, C, and E, there was a gradual decrease in the parameters over the storage period. For sample D, both parameters were similar at weeks 1 and 3 but then exhibited an abrupt change in week 5. Previous studies have shown that morphological change is not uniformly associated with age [24] and therefore, the different time points where the changes in physical parameters appears could be attributed to sample variations. There is a known donor variability that depends on factors such as age, sex, and environmental attributes [25,26] that is also well represented by the difference in the time dependent trends between the donors for both parameters, as shown in Figure 5 and Figure 6.

### 3.3. Classification of RBCs Using Logistic Regression

The extracted morphological parameters were used to train logistic regression to classify the cells according to their storage time. Logistic regression builds classifiers based on a linear combination of metrics from training data using the maximum likelihood method with logit link function. In order to efficiently train the classifiers, their performance was evaluated as a function of number of cells imaged using a binary classification between cells in weeks 1 and 5. The algorithms were built with randomly selected training data and then tested on the remaining dataset and repeated 10 times. The mean and the standard deviation of the classification accuracy for the different training data sizes are represented as error bar plots, shown below in Figure 7.

For all the samples, the accuracy of the algorithms increased with the number of the training data until the gain in the performance became insignificant. The improvement in the performance of the algorithms was comparable to the standard deviation of the accuracies across the trials when the number of training data images used in the algorithm was 4096. Therefore, the classifiers were trained with 5000 randomly selected cell images from each class while the performance was evaluated using the remaining dataset. The process was repeated 10 times to acquire the mean and the standard deviations of the classification accuracies.

### 3.4. Self-Trained Binary Classification: Week 1 vs. Week 5

The classification performances for the cells in week 1 vs week 5 using the algorithms trained on the same population are shown in Table 3. When the algorithm was trained with cells from Sample A, the classifier showed the highest performance with 85.6% accuracy, where 90.8% and 81.5% of the cells from Sample A were correctly identified as cells from week 1 and 5, respectively.

### 3.5. All Sample Binary Classification: Week 1 vs. Week 5

As a further examination, when the algorithm was trained with cells randomly selected from all the samples, its performance across the different donors is shown in Table 4.

The classifiers trained with the cells from multiple samples were able to distinguish the storage time point of the cells from the different donors with 69.9% accuracy, where 76.1% and 65.1% of the cells were classified correctly as cells from week 1 and 5, respectively.

## 4. Discussion

Throughout this study, holographic cytometry was used to image a large number of RBCs from stored blood units at the individual cell level to identify morphological changes over the storage period. By imaging cells flowing through the channels of the microfluidic chips, the throughput of holographic cytometry is no longer limited by the manual translation of sample stages. The use of multiple parallel channels further increased throughput while the use of stroboscopic illumination allowed for a high flow rate without image streaking. At the maximum throughput of the current setup, it took less than one second to image hundreds of cells, which is the same number of cells from our previous experiments [15]. Therefore, holographic cytometry can efficiently expand the utility of QPM by imaging a significantly greater number of cells relative to traditional QPM imaging systems.

One of the advantages of imaging a large number of cells is that the sensitivity of the system can be characterized at a much higher level of precision, enabling evaluation of rare conditions such as blood infections with low parasitemia percentages [27]. The diagnostic capability of a system will be limited by the lowest number of rare target cells that can be detected within a population. Therefore, by imaging a greater number of cells, the clinical utility of the system as a screening tool can be defined based on classification performance without the lower bound restriction arising from the total number of acquired cell images.

Out of the 25 morphological parameters extracted from the images, OV and area showed a consistent decrease over the storage time for all the samples except Sample A, at week 5, as shown by the Figure 5 and Figure 6. A possible explanation for the rise in the number of cells with decreased area over the storage period could be due to the previously reported transformation of discocytes to morphologically altered RBCs with smaller projected surface areas such as echinocytes, spheroechinocytes, and spherocytes [28,29,30]. One of these studies showed that morphologically altered RBCs accounted for 4.9% of the blood population on day 3, which increased to 23.6% by day 42 [29]. Therefore, the fairly consistent decrease in area over the storage time observed here can be a result of the echino-spherocytic shift of RBCs during storage. Previous studies have also shown a decrease in MCHC (mean corpuscular hemoglobin concentration) for stored RBCs. In the QPM measurements, this corresponds to a net decrease in the refractive index of the RBCs relative to the high RI medium, which would result in a decreased OV [28,31], as seen here. In a previous study using QPM, no correlation was observed between the change in OV with storage time and hemoglobin concentration, suggesting that the morphological changes observed here offer new insight compared with bulk measurements like hemolysis. It should also be noted that the physical parameters of the cells from Sample A did not continue to decrease at the week 5 storage time, unlike the cells from Samples B–E that showed steady changes over the storage time. This exception to the trend may have arisen from variations in sample handling or differences in the characteristics between the donors. In order to fully understand this, RBCs from a greater number of different donors must be examined.

Logistic regression, trained by the large data set of extracted parameters, is used to classify the cells according to the storage time points. Learning curves, shown in Figure 7, compare the classification accuracies relative to the training data set size to determine the optimal number of cell images to use. For all samples, the increase realized in classification accuracies is comparable to the standard deviation seen for the variation in performance across trials when there are ~4096 cell images used as training data. Therefore, a threshold of 5000 randomly selected cells from each sample are viewed as sufficient to build the classification algorithms, and the remaining cell imaging data are used as the test set.

Another advantage that can be realized by exploiting the large number of cell images that are acquired with holographic cytology can be seen by noting the number of images required to construct the training data. When logistic regression is trained with a low number of cells, the trained algorithms cannot distinguish images in the test data at optimal performance and therefore it can be assumed that the variability within the total population is not completely captured by a smaller training data set. The number of data points required to capture the characteristics of the total population vary depending on the type of the analyzed sample and the selected diagnostic criteria. For the classification of QPM images of cells in stored blood units based on logistic regression, the analysis here showed that the number of training data should be greater than 5000 cells, which would have been very difficult to accomplish using QPM with conventional scanning approaches such as using manual or automated translation stages.

The performances of algorithms for binary classification between week 1 and 5 are shown in Table 3. These algorithms were trained and tested by analyzing only cells from within the same sample. For the algorithm trained and tested with only the cells from Sample A, the highest accuracy in classifying the cells according to storage time was achieved at 85.6% accuracy. The lowest performance was for the algorithm which was trained only on images of cells from Sample E, with 78.2% accuracy. Given that Sample A exhibits the only outlier in the trends of OV and cell area, it is important to note that 23 other morphological parameters, extracted from the quantitative phase images, were also used to train the algorithms, allowing high classification accuracies. Thus, even though OV and cell area are presented in detail here, all of the remaining morphological parameters showed significant differences over the storage time. This illustrates the wealth of information that can be obtained from QPM images.

For algorithms that were trained using randomly selected images of cells across all the samples to build a binary classifier (as shown in Table 4), the overall classification accuracy decreased to 69.9%, which is a decrease that should be expected. The decrease in the performance of the algorithm relative to those trained only within the same population can be attributed to donor variability [25,26]. As can be seen in Figure 5 and Figure 6, the RBCs from different donors had varying morphological parameter distributions as well as distinctive changes over time that describe the variation in storage lesion. While some general trends in storage lesion can be expected to hold for all potential donors, it stands to reason that it may manifest with small variations depending on the source of the blood unit as well as its handling. Therefore, it is reasonable to expect that the performance of the universal classifier would decrease relative to those trained on individual samples. In order to fully characterize the range of morphological changes that can appear due to storage lesion, broader studies with RBCs from a greater number of different donors should be undertaken.

One possible source of variation in the observed trends across samples is the fact that each unit of RBCs is comprised of a continuum of cells of various ages from those at day 1 in the circulation to those that are at the end of the circulatory life [30]. Hence, the age distribution of cells imaged at one specific storage time point will inevitably have an overlap with the other time points examined here and therefore some cross-classification between storage time points by the algorithms is unavoidable. In the future, holographic cytometry could be used to image RBCs that have been fractionated by age to acquire information from cells with narrower age distribution, thereby possibly increasing the overall performance of the algorithms and improving our understanding of storage lesion. In addition, the results of holographic cytometry could be combined with other approaches to enhance the ability to detect blood doping. For example, the discovery of the RNA species in RBCs [32] enabled the identification of the transcriptional changes during RBC storage [33], which may be used together to enhance our ability to identify stored RBCs.

## 5. Conclusions

In this study, we employed holographic cytometry to image large numbers of individual cells at high throughput. For application to storage lesion, the approach provides data that better represent the variation within blood units than randomly sampling a handful of cells alone. By characterizing a greater number of cells at the individual level than previous studies, holographic cytometry was able to identify consistent morphological changes in RBCs over the storage period with trends that agree with the previous literature’s findings. In the future, by imaging samples from more subjects with diverse demographics, donor variability can be evaluated for establishing a cell-based predictor of transfusion yield. The potential of holographic cytometry as a screening tool was demonstrated by the classification performance of the logistic regression algorithms trained with morphological traits extracted from individual cell images, rather than from whole cell images. This distillation step enables the examination of a broader population than would be enabled by direct evaluation of the cell images, where the volume of data may pose an obstacle for efficient training and application of algorithms. The classification algorithms showed promising accuracies that could be improved by narrowing the distribution of the training data using demographic information such as age, sex, and health history, among others.

## Figures and Tables

**Figure 1 cells-10-02455-f001:**
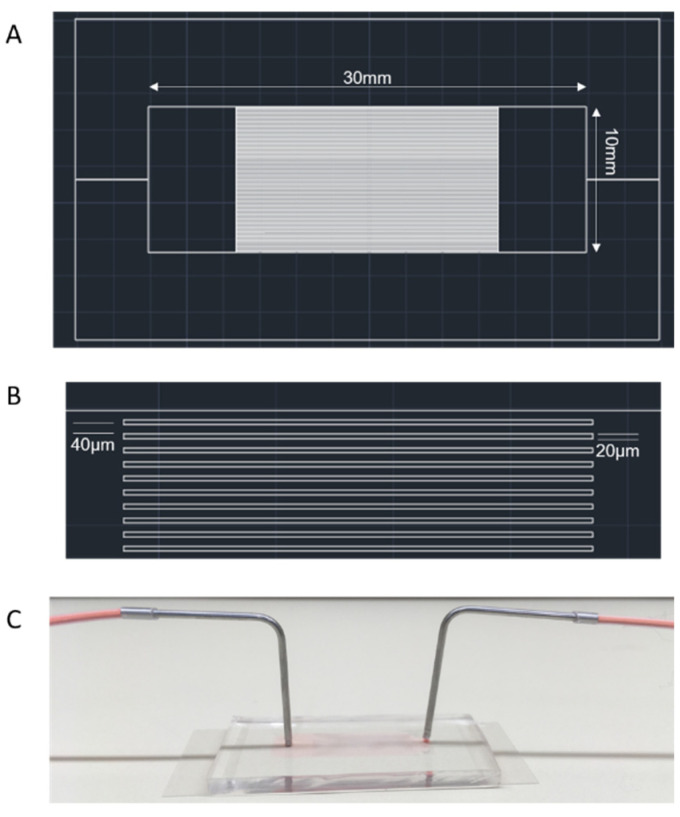
(**A**) Overall mask design for the PDMS channels. (**B**) Magnified mask design showing the dimensions of the channels. (**C**) PDMS channel.

**Figure 2 cells-10-02455-f002:**
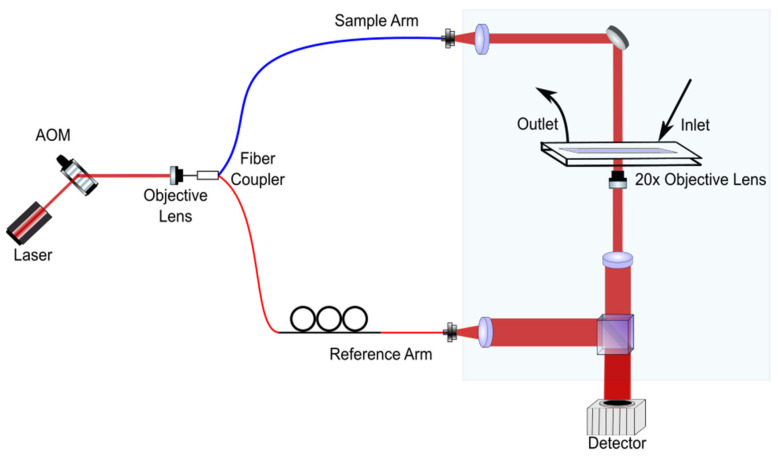
Holographic cytometry for high-throughput imaging.

**Figure 3 cells-10-02455-f003:**
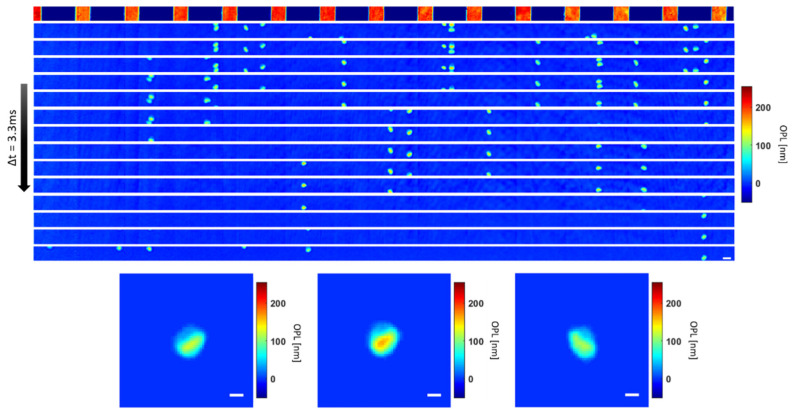
Top: Flow sequence with channels shown at the top subtracted as a background. Scale bar = 10 μm. Bottom: Selected red blood cells from the flow sequence on top. Scale bar = 5 μm (Appendix A).

**Figure 4 cells-10-02455-f004:**
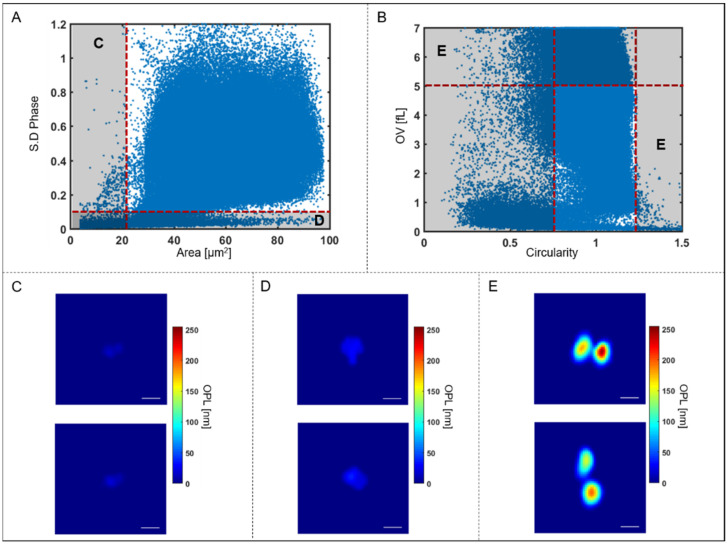
Illustration of outlier rejection based on QPI parameters. (**A**) Scatter plot showing standard deviation of phase and area of all objects imaged in a typical flow sequence before quality check. (**B**) Scatter plot showing OV and circularity of all objects in the sequence. Segmented objects with area < 21 μm^2^, (ex. (**C**)), shown on the scatter plot (**A**) by the red vertical dotted line and objects with standard deviation < 0.09, (ex. (**D**)), shown by the red horizontal dotted line are excluded from further analysis. Multiple-connected cells, i.e., clumps with OV > 5 fL and circularity <0.85 or >1.23, ex. (**E**), shown on the scatter plot in (**B**) by the red dotted lines are also excluded from further analysis. Scale bar = 5 μm.

**Figure 5 cells-10-02455-f005:**
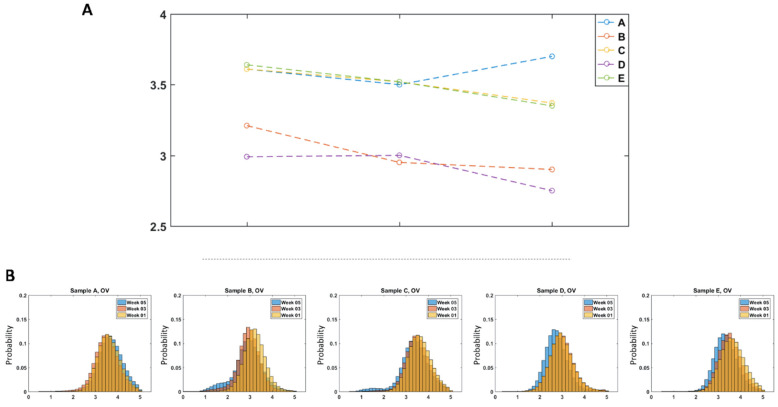
(**A**) Line plots of optical volume for all samples over the storage time (**B**) Normalized histogram of optical volume of cells over the storage period for Sample A–E.

**Figure 6 cells-10-02455-f006:**
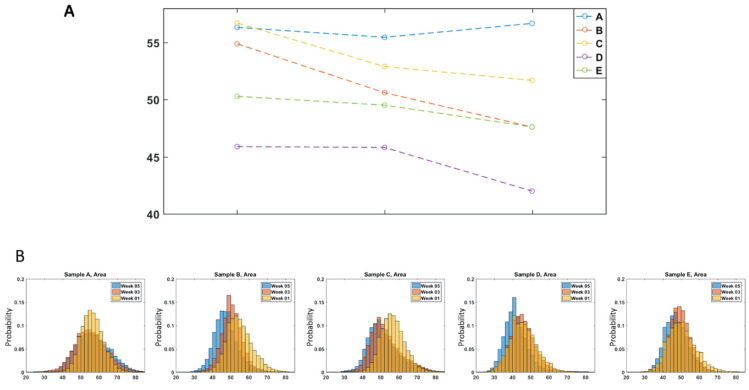
(**A**) Line plots of area for all samples over the storage time, (**B**) Normalized histogram of area of cells over the storage period for Sample A–E.

**Figure 7 cells-10-02455-f007:**
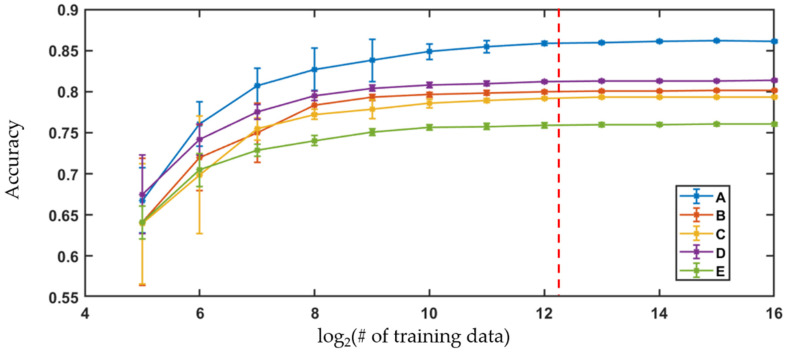
Learning curves showing performance of logistic regression versus number of training data for Sample A–E. Dotted red line = 5000 training data.

**Table 1 cells-10-02455-t001:** Basic information of blood donors.

	Sex	Year of Birth
Sample A	Male	1985
Sample B	Female	2000
Sample C	Male	1988
Sample D	Female	1964
Sample E	Male	1955

**Table 2 cells-10-02455-t002:** Total number of segmented RBCs.

	Sample A	Sample B	Sample C	Sample D	Sample E
Day 01	653,835	585,142	742,151	342,148	144,187
Day 15	788,569	1,387,083	704,062	117,985	135,920
Day 29	813,324	1,046,341	845,081	709,039	422,482

**Table 3 cells-10-02455-t003:** Binary classification of cells from week 1 and 5 using logistic regression trained and tested with cells from the same sample.

Classification Performance Avg ± std [%]	Sample A	Sample B	Sample C	Sample D	Sample E
Week 01	90.8 ± 0.3	82.0 ± 0.4	83.2 ± 0.4	77.3 ± 0.3	71.6 ± 0.2
Week 05	81.5 ± 0.3	78.1 ± 0.3	75.3 ± 0.4	85.1 ± 0.3	80.4 ± 0.2
Accuracy	85.6 ± 0.1	79.5 ± 0.1	79.0 ± 0.1	82.6 ± 0.1	78.2 ± 0.1

**Table 4 cells-10-02455-t004:** Binary classification of cells from week 1 and 5 using logistic regression trained and tested with cells from all samples.

Classification Performance Avg ± std [%]	Week 01	Week 05
Week 01	76.1 ± 0.4	23.9 ± 0.4
Week 05	34.9 ± 0.6	65.1 ± 0.6

## Data Availability

The data presented in this study are available on request from the corresponding author. The data are not publicly available due to the large volume of imaging data.

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
