# Peer review of "Single Cell Analysis of Stored Red Blood Cells Using Ultra-High Throughput Holographic Cytometry"

_cells, 2021, doi:10.3390/cells10092455_

Round 1
Reviewer 1 Report
The authors present a novel approach to image large number of individual cells at high throughput. The methodology part is described in detail. The authors found differences between the samples with different storage time , which were also used by machine learning algorithm.
As far as I understood, the aim is to find new (objective, less laborious) methodology to evaluate storage lesion. The machine learning algorithm is capable (in majority of tested samples) of identification of storage time (week 1 vs week 5). Nevertheless, as the storage time is known information, I am curious how do the results correlate to sample quality (or vice versa, storage lesion)? Did the authors compare the results with other parameters (hemolysis?), which this methodology may enrich/substitute to evaluate storage lesion?
As the cells with extreme values of OV/area were excluded (as described in Methods) and these two values were observed to change with storage time, did the authors observe differences in proportion of cells that were excluded from the analysis?
The authors speculate that sample A might have differed because of variation of sample handling or other parameters. Did they uncover any differences? Were all the samples drawed and proccessed the same way?
Is the number of samples processed (n=5) adequate for statements about continuous changes during storage? Has any statistical test been applied?
Minor comments/suggestion:
In abstract, I suggest to include more detailed information of what was performed (how many samples, minimum cell number for machine learning algorithm etc).
In introduction - flow cytometry is capable to detect much more than 14 parameters (reference no.2 - published in 2012). Either I would skip intro about fluorescent flow cytometry or correct the statement.
Shotcut PDMS is not explained.
Author Response
Response to Reviewer 1
The authors present a novel approach to image large number of individual cells at high throughput. The methodology part is described in detail. The authors found differences between the samples with different storage time, which were also used by machine learning algorithm.
Response: We thank the reviewer for their careful reading of our manuscript and identifying the novelty of our approach.
As far as I understood, the aim is to find new (objective, less laborious) methodology to evaluate storage lesion. The machine learning algorithm is capable (in majority of tested samples) of identification of storage time (week 1 vs week 5). Nevertheless, as the storage time is known information, I am curious how do the results correlate to sample quality (or vice versa, storage lesion)? Did the authors compare the results with other parameters (hemolysis?), which this methodology may enrich/substitute to evaluate storage lesion?
Response: We thank the reviewer for identifying the study goal which is to show that the machine learning analysis using individual RBC images can identify storage time. The reviewer seeks to understand how sample quality degrades with storage time. Typically, this is measured by hemoglobin analysis (such as hemolysis). As expected from the routine blood storage conditions, we did not note observable evidence of hemolysis. Previously, we have performed microRNA profiling to determine the transcriptional response. However, that approach was performed in bulk samples and on different subjects (Yang et al, Ref 33). Therefore, it is not possible to make comparison with the single cell analysis results shown here. Although methods like ektacytometry have sought to study cell deformability due to storage lesion, the approach is not generally well adapted to high throughput. The true advance in our approach is to translate a single cell analysis method to high throughput. We have examined storage lesion with QPI previously (Ref 20) but were limited to significantly smaller numbers of cells. In this previous study, we examined parameters such as Optical Volume and found there was no correlation with changes in hemoglobin concentration over storage time. The revised paper includes a citation to this paper in the discussion, 3rd paragraph (line 316-320).
“Previous studies have also shown a decrease in MCHC (mean corpuscular hemoglobin concentration) for stored RBCs. In the QPM measurements, this corresponds to a net decrease in refractive index of the RBCs relative to the high RI medium, which would result in a decreased OV [28,31], as seen here. In a previous study using QPM [21], no correlation was observed between the change in OV with storage and hemoglobin concentration, suggesting that the observed morphological changes offer new insight compared to bulk measurements like hemolysis.”
As the cells with extreme values of OV/area were excluded (as described in Methods) and these two values were observed to change with storage time, did the authors observe differences in proportion of cells that were excluded from the analysis?
Response: The excluded objects were either too small to be cells or too large to be individual cells. We have revised the text to clarify this further (line 179-180,184-5). These objects were excluded from analysis and thus the number of these objects was not tracked.
The authors speculate that sample A might have differed because of variation of sample handling or other parameters. Did they uncover any differences? Were all the samples drawed and proccessed the same way?
Response: Sample A was the first one examined in this study and thus it was handled more often than the others as it was removed from storage to draw aliquots for analysis. Although we were more careful in handling the subsequent samples, we were unable to identify the difference and thus we feel the statement in the paper is the best way to address our suspicions (lines 320-321)
Is the number of samples processed (n=5) adequate for statements about continuous changes during storage? Has any statistical test been applied?
Response: The goal of this paper is to introduce a new method for characterizing stored blood samples based on QPM morphology measurements. The use of 5 samples was chosen to characterize the variability of the parameters across samples. The manuscript clearly describes the limitations due to sample size and suggests means to conduct future experiments (lines 324, 377, 392). The publication of our manuscript will allow us and other investigators to continue to improve and validate these findings in a much larger number of samples and independent manner.
Minor comments/suggestion:
In abstract, I suggest to include more detailed information of what was performed (how many samples, minimum cell number for machine learning algorithm etc).
Response: We have updated the abstract to discuss number of samples and cell numbers (lines 15-16, 22-23)
In introduction - flow cytometry is capable to detect much more than 14 parameters (reference no.2 - published in 2012). Either I would skip intro about fluorescent flow cytometry or correct the statement.
Response: We have revised this statement to remove the reference to 14 parameters. “…flow cytometry has evolved into a sophisticated modality that can simultaneously a wide range of optical and fluorescence parameters”
Shotcut PDMS is not explained.
Response: We have explained this acronym on line 96.

Reviewer 2 Report
In this study, the authors developed a high-throughput quantitative phase microscopy (QPM) by combining microfluidics and QPM. This new method allows rapid and high-throughput quantification of multiple cellular morphological characteristics.
The authors applied this method to study the morphological changes of RBC during storage and tried to train logistic regression models for RBC storage time point classification. Based on the results, good classifications were achieved while using the sample-specific models. As stated in the discussion, it is not surprising that gender and age would have effects on the parameters used for classification. Building more datasets fractioned by age and gender might be helpful. Further studies using this platform would be beneficial for blood doping detection in sports.
Overall, this platform could be interesting to many other readers in need of high-throughput rapid capture and analysis of cell morphological characteristics. Fields such as cell and cancer biology could benefit from this platform. So, I recommend this manuscript to be considered for publication in Cells after minor revision.
Minor suggestions:
1、According to the authors, they extracted a set of morphological parameters of millions of RBCs to assess the RBCs storage lesion which reduced the viability of RBCs, but they did not do any other detection to demonstrate the lesion of cells over time. Can the authors provide more analysis or improvement?
2、In the methods part, the detail of the fabrication of the PDMS channels should be described. In the results part, although, the author briefly described the process of training the classifier, more detail should be shown in the methods part.
3、In order to make the results shown clearly, the description should be more clear, for example in the caption of Figure 4.
4、In Figure 4, contour plots could be a better substitute for the scatter plots (Figure 4 A&B) when displaying data with high event counts.
5、Please modify the format of the numbers and the units in the manuscript. There is a space between the number and the unit.
Author Response
Response to Reviewer 2
In this study, the authors developed a high-throughput quantitative phase microscopy (QPM) by combining microfluidics and QPM. This new method allows rapid and high-throughput quantification of multiple cellular morphological characteristics.
Response: We thank the reviewer for identifying the capabilities of our new method.
The authors applied this method to study the morphological changes of RBC during storage and tried to train logistic regression models for RBC storage time point classification. Based on the results, good classifications were achieved while using the sample-specific models. As stated in the discussion, it is not surprising that gender and age would have effects on the parameters used for classification. Building more datasets fractioned by age and gender might be helpful. Further studies using this platform would be beneficial for blood doping detection in sports.
Response: We again thank the reviewer for realizing that sample variability is to be expected and that further studies are needed for broader application.
Overall, this platform could be interesting to many other readers in need of high-throughput rapid capture and analysis of cell morphological characteristics. Fields such as cell and cancer biology could benefit from this platform. So, I recommend this manuscript to be considered for publication in Cells after minor revision.
Response: We thank the reviewer for recommend publication after revisions that will strengthen the manuscript.
Minor suggestions:
1、According to the authors, they extracted a set of morphological parameters of millions of RBCs to assess the RBCs storage lesion which reduced the viability of RBCs, but they did not do any other detection to demonstrate the lesion of cells over time. Can the authors provide more analysis or improvement?
Response: We have previously conducted similar experiments in stored blood (ref 21) and showed comparisons of QPI with traditional methods as well as comparing QPI to chemically aged cells using glutaraldehyde. The storage lesion is fairly consistent over samples. We have added a statement on this on lines 191-192
2、In the methods part, the detail of the fabrication of the PDMS channels should be described. In the results part, although, the author briefly described the process of training the classifier, more detail should be shown in the methods part.
Response: We have added a reference to the fabrication of PDMS channels. This is a fairly standard technique at this point and a detailed description here would impact the flow of the paper. We have added a bit more explanation of the Logistic Regression algorithm used her on lines 239-241.
3、In order to make the results shown clearly, the description should be more clear, for example in the caption of Figure 4.
Response: We have added a title for Figure 4 to better indicate the content of the figure.
4、In Figure 4, contour plots could be a better substitute for the scatter plots (Figure 4 A&B) when displaying data with high event counts.
Response: We seek for these data to resemble flow cytometry plots. While we agree that contour plots would be more useful for analyzing these plots, here we seek just to show the general trends for discussing outlier removal. The histogram plots in the subsequent figures provide a better means for analysis.
5、Please modify the format of the numbers and the units in the manuscript. There is a space between the number and the unit.
Response: We have made this revision.
Round 2
Reviewer 1 Report
The authors commented and answered the questions. The abstract was significantly improved and includes now concrete data. I suggest to perform minor language correction of the following sentence: "Samples from five blood donors were examined over 100,000 cells examined for each at three time points." (sounds as mistyping error with repeating verb). Otherwise I do not have any further questions nor comments.